# Oxygen Reduction Behavior of HDH TiH$_2$ Powder during Dehydrogenation Reaction

**Ki Beom Park [1,2], Jaeho Choi [2], Tae-Wook Na [1]** **, Jang-Won Kang [1], Kwangsuk Park [1,\*] and Hyung-Ki Park [1,\*]**

[1] Gangwon Regional Division, Korea Institute of Industrial Technology, Gangneung 25440, Korea; hope92430@kitech.re.kr (K.B.P.); arkasa86@kitech.re.kr (T.-W.N.); jwkang@kitech.re.kr (J.-W.K.)

[2] Department of Advanced Metal and Materials Engineering, Gangneung-Wonju National University, Gangneung 25457, Korea; cjh@gwnu.ac.kr

[\*] Correspondence: kpark63@kitech.re.kr (K.P.); mse03@kitech.re.kr (H.-K.P.)

**Abstract:** In this study, oxygen reduction behavior of TiH$_2$ powders during dehydrogenation process was investigated based on thermodynamics. During the hydrogenation–dehydrogenation (HDH) method to fabricate Ti powder, TiH$_2$ was formed from a Ti sponge through hydrogenation annealing, and was easily pulverized even by ball milling due to its brittle nature. The ball milling process caused an increase in the oxygen concentration from 0.133 to 0.282 wt %, and transmission electron microscopy and X-ray photoelectron Spectroscopy results demonstrated that the formation of oxide layers such as TiO and TiO$_2$ formed on the surface of the TiH$_2$ powder resulted in the higher oxygen content. Dehydrogenation, which is the process originally conducted to eliminate hydrogen from TiH$_2$, was used to remove and/or reduce oxygen, resulting in the reduction of the oxygen concentration from 0.282 to 0.216 wt %. Thermodynamic calculations confirmed the possibility of oxygen reduction by atomic hydrogen but molecular hydrogen has no function for the oxygen reduction. Glow discharge mass spectrometry (GD-MS) analysis, which checks H$_2$O flow as an evidence of the oxygen reduction by hydrogen, supported the fact that the atomic hydrogen formed during the dehydrogenation process is able to play a critical role in decreasing the oxygen content.

**Keywords:** titanium; powder; hydrogenation–dehydrogenation; oxygen reduction

## 1. Introduction

Ti and Ti alloys have high strength and low density; therefore, they are widely used in aerospace industries [1,2]. In addition, the passivation characteristic of Ti and its alloys provides good corrosion resistance, and allows them to be used in a variety of chemical and biomedical industries [3–5]. However, due to the high oxidation driving force and low thermal conductivity of Ti, it is not easy to fabricate parts through manufacturing processes such as cutting and forming [6]. Therefore, powder metallurgy would be promising way to make parts with Ti and its alloys [7,8].

Lots of researches on powder fabrication processes of Ti, such as the hydrogenation–dehydrogenation (HDH) process [9–11] and electrode induction gas atomization (EIGA) have been carried out [12,13]. Recently, with the development of the additive manufacturing technology, the need for an advanced method to make spherical Ti powders is being increased [12–14]. EIGA could produce spherical Ti powders with high quality but the manufacturing cost is pretty high due to the preparation of the Ti electrode and low yield for the specific range of Ti powders. On the other hand, the HDH method, where hydrogen embrittlement of Ti is a key factor, is able to fabricate Ti powders with low cost, but the quality of the powders would not be satisfied in terms of oxygen contamination which is induced during a milling process [15]. The HDH method is a technique that can produce a

large amount of powders in an inexpensive method because the powders are fabricated by crushing after hydrogenating Ti [9].

The Ti powder fabricated by EIGA has a spherical shape, whereas the Ti powder fabricated by the HDH method has an irregular shape because it is fabricated by grinding [9,10]. Thus, the powder flowability of the HDH Ti powder is much lower than that of the EIGA Ti powder. In addition, in the case of HDH Ti powder, the oxygen concentration is higher than EIGA Ti powder due to the contamination during milling [11], and it is necessary to develop a technology for reducing the oxygen concentration.

Ti could absorb hydrogen during annealing under a suitable hydrogen condition, resulting in $TiH_2$ hydride. Due to the brittle nature of $TiH_2$, $TiH_2$ could be easily broken down even by a ball milling process and turned into metallic Ti powders through the annealing in vacuum to remove hydrogen from the $TiH_2$. High oxygen affinity of Ti causes an increase in oxygen concentration of the Ti powders during the milling process so the excess oxygen should be eliminated as it hurts toughness of Ti when the oxygen concentration exceeds 0.3 wt % [16], suggesting that the oxygen content in the Ti powder has to be controlled.

Therefore, in this study, we introduce a new approach in reducing the oxygen content in Ti and confirm its effect on oxygen reduction by analyzing the change of the oxygen concentration with the milling process and dehydrogenation of hydrogenated $TiH_2$ as well as thermodynamic calculations. A glow discharge mass spectrometry (GD-MS) was also used to support that the oxygen reduction could occur during the dehydrogenation process.

## 2. Experimental Procedures

Grade 1 commercially pure Ti sponges were used as an initial sample in this study. In order to make $TiH_2$ through hydrogenation, the sponge was heated to 10 °C/min in a 100% $H_2$ atmosphere, and annealed for 2 h after reaching 600 °C. After hydrogenation, $TiH_2$ could be easily crushed due to its brittle nature, and was first pulverized into particles of 1 mm or less in size using a jaw crusher. Thereafter, the particles were placed in a 1000 cc stainless steel container and pulverized into powders using a 5 mm diameter tungsten carbide balls as a milling medium. The powder-to-ball ratio was 1:5 by weight. The steel container was purged by high purity argon to prevent oxygen contamination and the powders were ball-milled for 1 h at a rotation speed of 250 rpm. After that, the powders were sieved by using meshes ranging in size from 25 to 63 μm. To remove hydrogen from the $TiH_2$ powders, dehydrogenation annealing was carried out at 600 °C for 2 h with a heating rate of 10 °C/min in a high vacuum (~$10^{-5}$ torr) atmosphere.

The morphology of the Ti powders fabricated through HDH reaction was observed using a field emission-scanning electron microscope (FE-SEM) (QUANTA FEG 250, FEI, Hillsboro, OR, USA). Phase analyses of the samples were carried out using an X-ray diffractometer (XRD) with Cu K$\alpha$ radiation in the 2θ range of 30–80°. The powder size distribution of the hydrogenated $TiH_2$ powders and the dehydrogenated Ti powders were analyzed using a powder size analyzer (Mastersizer 3000, Malvern Panalytical, Malvern, UK). After ball milling, the surface of the $TiH_2$ powder was analyzed using a field emission-transmission electron microscope (FE-TEM) (JEM-2100F, JEOL, Akishima, Japan) to confirm the formation of oxide layer on the surface. A sample for the TEM analysis was prepared by a focused ion beam instrument (Helios NanoLab, Hillsboro, OR, USA). In addition, the composition of the oxide layer formed on the surface was analyzed using an X-ray photoelectron spectroscopy (K Alpha+, Thermo Fisher Scientific, Waltham, MA, USA).

In order to confirm the possibility of oxygen reduction during the dehydrogenation process, the oxidation-reduction behavior of Ti oxide was analyzed based on the thermodynamics using Thermo-Calc with an SSUB5 database. In addition, the oxygen reduction behavior in the dehydrogenation process was analyzed by the amount of $H_2O$ released according to time and temperature using a GD-MS (STA 449 Jupiter, Netzsch, Berlin, Germany). The GD-MS experiment was carried out in a high purity argon atmosphere with temperature up to 1000 °C at a heating rate

of 10 °C/min. Oxygen concentrations before and after dehydrogenation were analyzed using an inert gas fusion infrared absorption method employing an oxygen analyzer (LECO, 736 series, LECO, St. Joseph, MA, USA) with a graphite crucible. Oxygen concentration was obtained by averaging five measurements.

## 3. Results and Discussion

Figure 1a,b shows the morphology of $TiH_2$ and Ti powders fabricated by the HDH method, demonstrating that the powders have an irregular shape as a result of the brittle fracture. Ti thermodynamically reacts with hydrogen under the proper annealing condition in a hydrogen atmosphere [17–19]. As a form of metal hydride, $TiH_2$ would experience hydrogen embrittlement, enabling it to be pulverized by simple milling process. As shown in Figure 1a,b, the powders show a smooth surface instead of the dimple morphology as an evidence of the brittle fracture, which is very similar with the previous results [17–19].

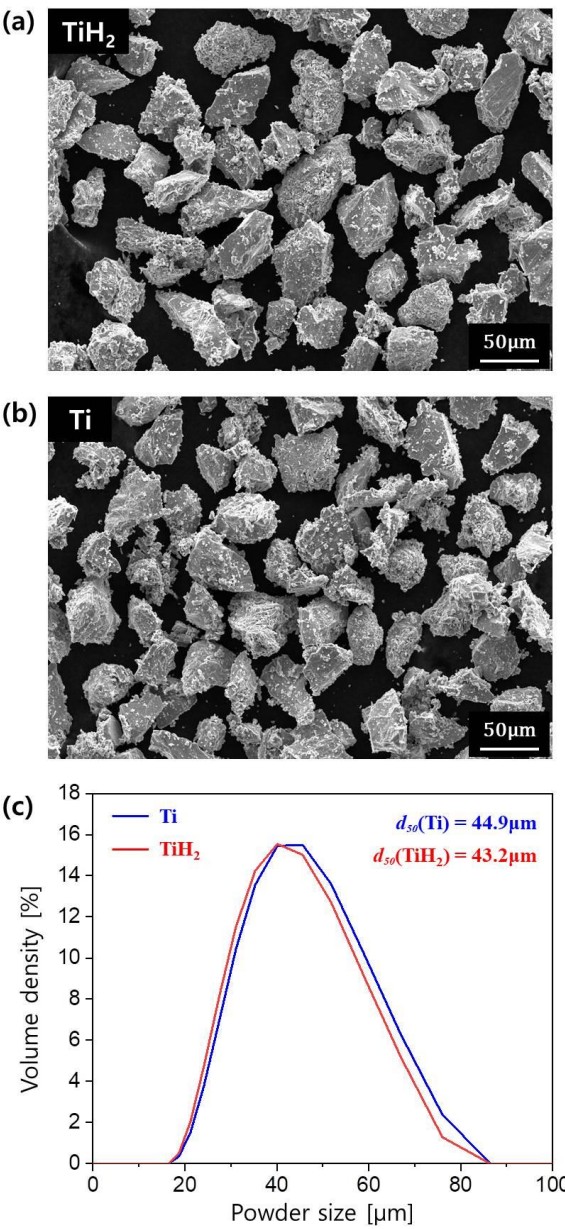

**Figure 1.** Morphology of (**a**) $TiH_2$ and (**b**) Ti powders fabricated by hydrogenation–dehydrogenation (HDH) process and (**c**) size distribution of $TiH_2$ and Ti powders.

In order to confirm the size change of the powders after removing hydrogen, the size distributions of the powders before and after the dehydrogenation were analyzed. Figure 1c shows the size distribution of the hydrogenated $TiH_2$ powders and the dehydrogenated Ti powders. Considering the size distribution of the powders in terms of $d_{10}$, $d_{50}$, and $d_{90}$, the sizes of $TiH_2$ powders were 28.6, 43.2, and 63.7 μm, and the ones of Ti powders were 29.7, 44.9 and 65.8 μm, respectively. The change in the powder size with the dehydrogenation process seemed not to occur, indicating that there is no effect on the size of the final metal powders after removing hydrogen.

Figure 2 shows the XRD patterns of the initial Ti sponge, the hydrogenated $TiH_2$ powder, and dehydrogenated Ti powder. Initially, the Ti sponge had α-Ti phase with a hexagonal close packed structure, and after the hydrogenation annealing, it was transformed to $TiH_2$ phase. Thereafter, brittle $TiH_2$ was pulverized to powders and hydrogen was removed through annealing in a vacuum atmosphere. The XRD result after the dehydrogenation annealing confirmed that all hydrogen was removed and the crystal structure was returned to the original α-Ti phase.

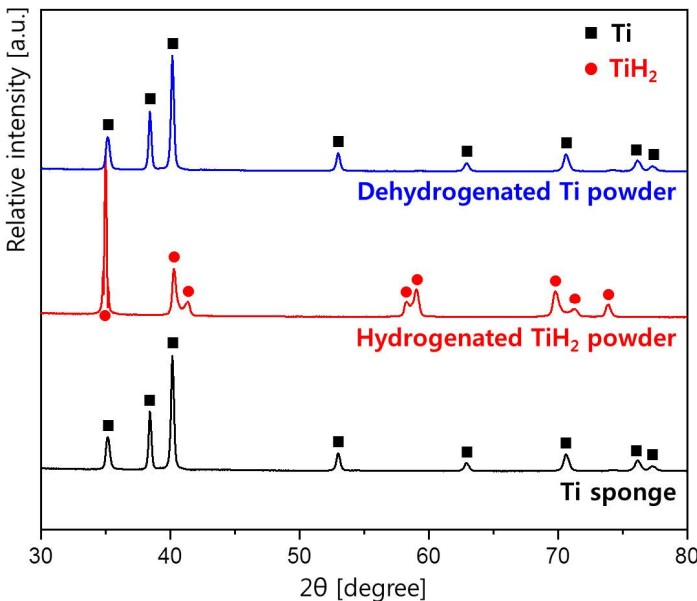

**Figure 2.** XRD patterns of the initial Ti sponge, the hydrogenated $TiH_2$ powder, and dehydrogenated Ti powder in the 2θ range from 30° to 80°.

The oxygen concentration in the powders as well as the size of the powders should be considered. As shown in Table 1, the ball milling seemed to cause the oxidation of the $TiH_2$ powders; the oxygen concentration was increased from 0.133 to 0.282 wt %. Even though the ball milling was conducted in argon atmosphere to prevent the $TiH_2$ powders from oxidizing, it seemed that the oxidation of the $TiH_2$ powders occurred. However, the oxygen content could be reduced during the dehydrogenation process in which the $TiH_2$ powders underwent the annealing in vacuum, resulting in the oxygen concentration of 0.216 wt %.

**Table 1.** Oxygen concentration of $TiH_2$ before and after milling and dehydrogenation Ti powders (wt %).

| Sample | Oxygen Concentration |
| --- | --- |
| $TiH_2$ (before milling) | 0.133 |
| $TiH_2$ (after milling) | 0.282 |
| Dehydrogenated Ti | 0.216 |

TEM analysis confirmed that the increase in oxygen concentration during the ball milling resulted from the formation of the oxide layer on the $TiH_2$ powders. Figure 3 is the TEM image showing the

surface microstructure of the ball-milled $TiH_2$ powder containing one layer formed on the surface. Energy dispersive X-ray spectroscopy analysis with a line scan mode revealed that the layer formed on the surface was an oxide film with the thickness of about 14 nm. The detailed analysis on the composition of the oxide layer such as an oxidation state was carried out through X-ray photoelectron spectroscopy (XPS).

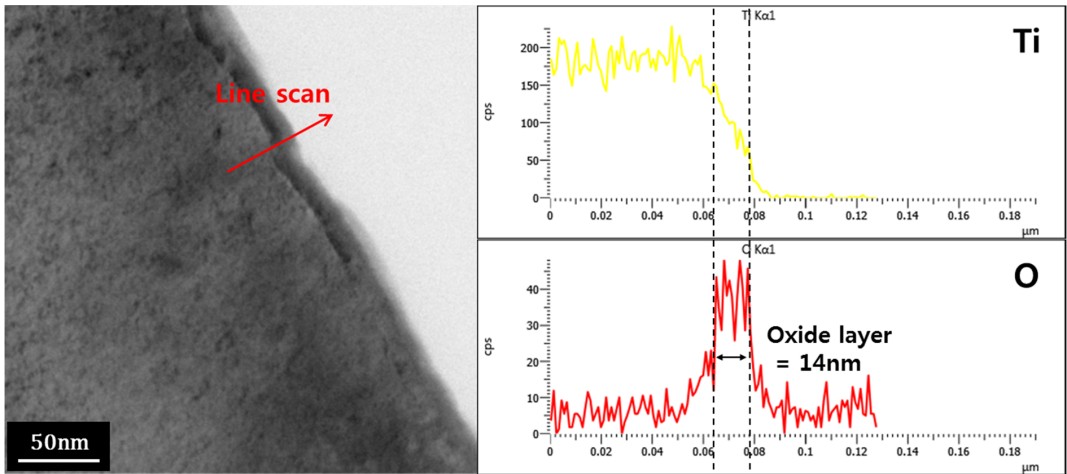

**Figure 3.** Surface microstructure of ball-milled $TiH_2$ observed by TEM. Line scan was performed to analyze the thickness of oxide layer.

As shown in Figure 4, the oxide layer had two oxidation states: $Ti^{4+}$ and $Ti^{2+}$. The strong peaks at 464 and 458 eV were observed in the spectrum. These peaks are attributed to Ti $2p_{1/2}$ and Ti $2p_{3/2}$, which is related with the $4^+$ oxidation state of Ti [20–22]. Minor peaks were observed at 455.5 and 453.5 eV, respectively, indicating $Ti^{2+}$ and $TiH_2$ states exist [21,22]. As $Ti^{4+}$ and $Ti^{2+}$ states indicate the presence of $TiO_2$ and TiO oxide, it can be concluded from the XRD result that $TiO_2$ oxide was mainly formed with a small amount of TiO oxide when the oxide layer was formed on the $TiH_2$ during the milling process.

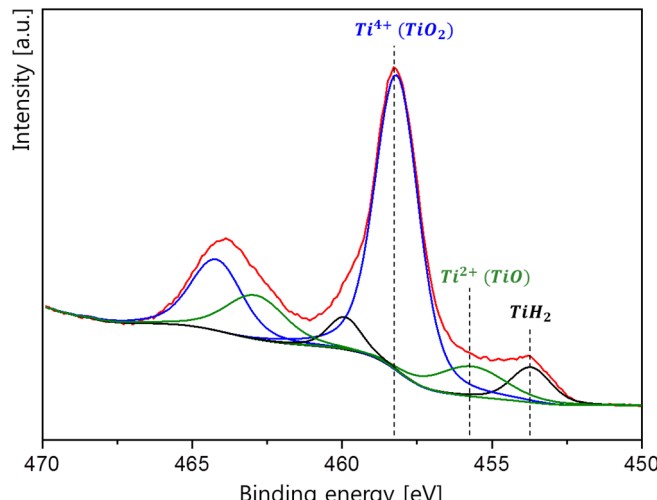

**Figure 4.** Surface analysis of the ball-milled $TiH_2$ powders by XPS.

The oxide layer, as the cause of the increased oxygen concentration, was reduced during the dehydrogenation reaction, as shown in Table 1, suggesting that a reduction reaction for the oxide layer occurred. It is commonly known that as Ti has high driving force for oxidation, it is hard to reduce Ti

oxides, especially through heat treatment in a vacuum or hydrogen atmosphere. As shown in Figure 5, illustrating a possible way to reduce the Ti oxides, atomic hydrogen was introduced as a tool for the reduction reaction.

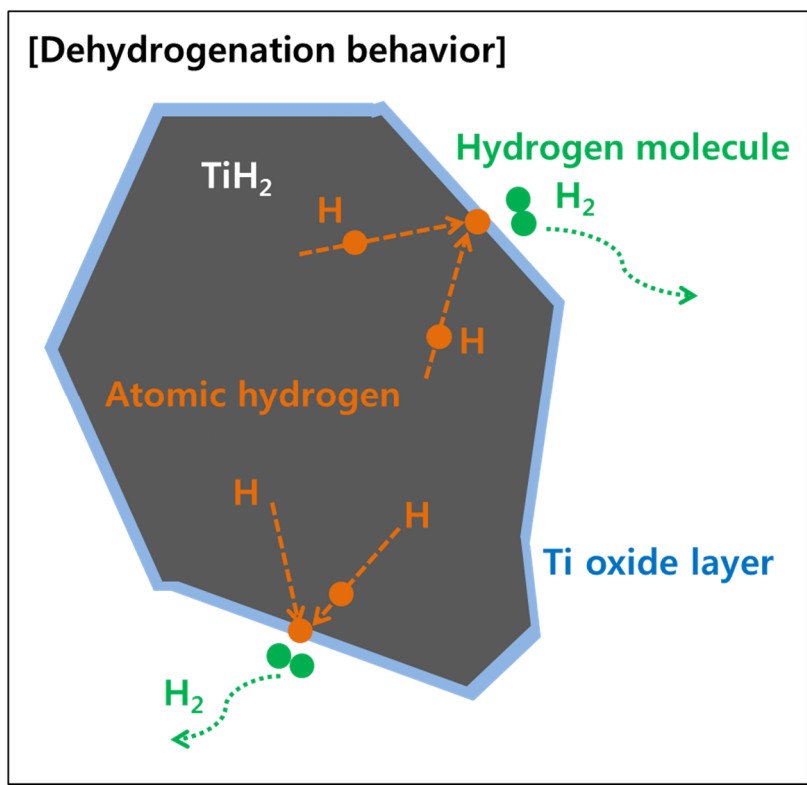

**Figure 5.** Schematic diagram illustrating the effect of atomic hydrogen removing oxide formed on the TiH$_2$ surface during the dehydrogenation process.

It is known that hydrogen in TiH$_2$ is dissolved in the form of atomic hydrogen, and hydrogen is released when the phase transformation occurs from TiH$_2$ to Ti under the dehydrogenation process. At this time, atomic hydrogen diffuses to the surface and is bonded to hydrogen molecules at the surface to be released as a gas state. Atomic hydrogen is known to have a much higher oxygen reduction driving force than hydrogen molecules [23]. If the oxide layer reacts with atomic hydrogen at the surface, the oxide layer might be reduced, resulting in a decrease in the oxygen concentration. The possibility for the reduction of the Ti oxide with the atomic hydrogen was proved through thermodynamic calculations.

Figure 6 is the thermodynamic calculation results for the reduction of TiO and TiO$_2$ oxides by hydrogen molecule and atomic hydrogen. The result shows that the standard Gibbs free energy change for the reduction reaction of the TiO and TiO$_2$ oxides with hydrogen molecule is positive; therefore, the reduction by hydrogen molecule is not possible. However, in the case of atomic hydrogen, it can be seen that the standard Gibbs free energy change is negative when reacting with the oxides to form Ti metal and H$_2$O, indicating that atomic hydrogen is able to reduce Ti oxides. From the results, it can be inferred that due to the presence of the atomic hydrogen released from the TiH$_2$ during the dehydrogenation process, controlling oxygen concentration could be achieved by removing hydrogen from the TiH$_2$.

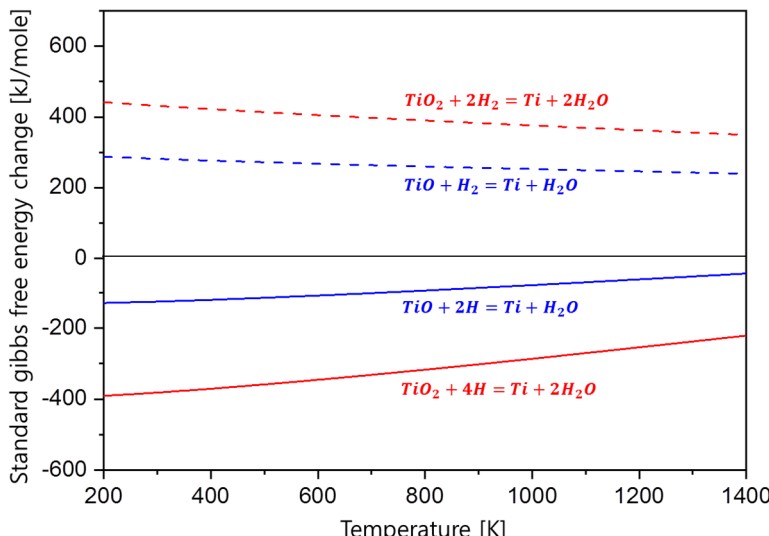

**Figure 6.** Standard Gibbs free energy changes in the reduction behavior of TiO and TiO$_2$ oxides with respect to hydrogen molecule and atomic hydrogen.

GD-MS measurement was applied to confirm whether the Ti oxides could be reduced by the atomic hydrogen. Figure 7 shows the GD-MS result, where the amount of H$_2$O flow with respect to the temperature was measured during the dehydrogenation process of TiH$_2$. H$_2$O flow is detected in GD-MS during heating metal materials when H$_2$O adsorbed on the metal surface evaporates or the oxide layer on the surface is reduced by hydrogen. Evaporation of H$_2$O adsorbed on the surface would occur near the boiling point of water and the generation of H$_2$O at higher temperatures would be caused by the reduction of the oxide layer.

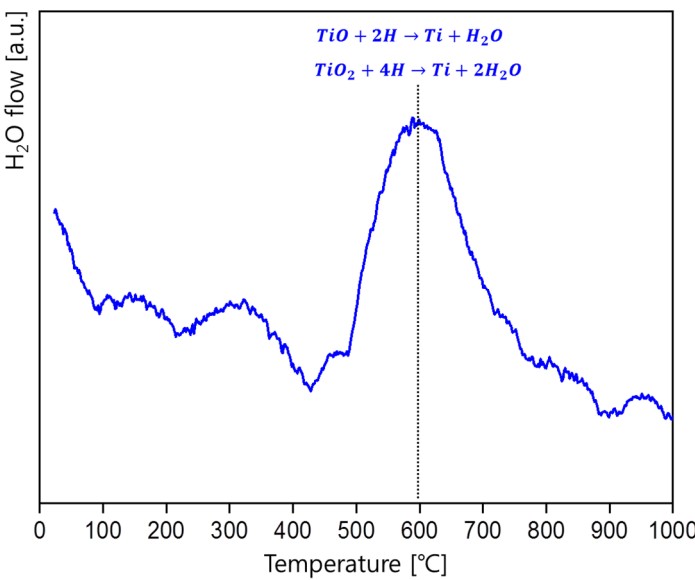

**Figure 7.** GD-MS result of TiH$_2$ during dehydrogenation process. The amount of H$_2$O flow was measured with respect to the temperature.

In general, when powders are heated, H$_2$O flow is observed up to 100 °C due to the adsorbed moisture on them. However, in the GD-MS result, the peaks showing the H$_2$O flow were observed when the temperature was up to 100 °C, and the other temperature around 500–600 °C, indicating that there is the reduction reaction around 500 °C.

For TiH$_2$, under the condition that the hydrogen partial pressure is low, hydrogen starts to be released with temperature as TiH$_2$ decomposes. It is reported that the hydrogen release usually occurs between about 400 and 500 °C [21], which is pretty similar with the temperature at which H$_2$O begins to be released, as shown in the GD-MS result. So, when Ti powders are fabricated through the HDH process, the oxygen concentration could be controlled by atomic hydrogen released from the TiH$_2$ powders even though oxidation occurs during the milling process.

## 4. Conclusions

In this study, the mechanism for the reduction of the oxygen on the TiH$_2$ powders during the dehydrogenation process was analyzed. The oxygen concentration of TiH$_2$ was increased after the ball milling process and TEM and XPS results showed that a TiO$_2$ and TiO oxide layer with a thickness of 14 nm was formed on the surface of the TiH$_2$ powders. The decrease in the oxygen concentration could be achieved when the atomic hydrogen was released on the surface during the dehydrogenation process. Thermodynamic calculations revealed that Ti oxides such as TiO$_2$ and TiO could be reduced if the atomic hydrogen was involved in the reaction of the oxygen reduction. As the evidence of the reduction process, the increase in H$_2$O flow that is an outcome of the reduction process was observed in the GD-MS result, indicating that the atomic hydrogen could reduce the Ti oxides.

**Author Contributions:** conceptualization, K.B.P. and K.P.; methodology, J.C.; investigation, T.-W.N. and J.-W.K.; writing—original draft preparation, H.-K.P.

**Funding:** This study was conducted with the support of the Korea Institute of Industrial Technology (Kitech EO-19-0007). Jaeho Choi acknowledges the support of Gangneung-Wonju National University.

**Conflicts of Interest:** The authors declare no conflict of interest.

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
