# Peer review of "Oxygen Reduction Behavior of HDH TiH2 Powder during Dehydrogenation Reaction"

_metals, doi:10.3390/met9111154_

Round 1

Reviewer 1 Report

Interesting paper.

In the abstract replace GD-MS by Glow Discharge Mass Spectrometry (GD-MS).

The references must avoid underlines.

Author Response

Replies to the comments of reviewer #1

Comment: In the abstract replace GD-MS by Glow Discharge Mass Spectrometry (GD-MS).

Answer:

- As mentioned by the reviewer, we replaced GD-MS by Glow discharge mass spectrometry (GD-MS) in the abstract

Comment: The references must avoid underlines.

Answer:

- As mentioned by the reviewer, we removed underlines in the references.

Reviewer 2 Report

The presented results were novel and original and gives additional knowledge for the Ti powders production. Despite off all benefits manuscript could be improved before publishing:

As presented results could be used for the Ti powders production, would be very useful to add some information or preliminary estimations in Introduction on process economy in comparison to existing Ti powders production methods. In other words is proposed approach not too expensive for example in comparison of Ti powders production using powder metallurgy processes? Figure 1 (a) presents Ti powders fabricated using HDH process. Would be useful to add TiH2 powders SEM views too. It would be also very useful to add XRD results Ti and TiH2 powders showing phase transformation during HDH process and to confirm that all the hydrogen was released in final Ti powders after HDH. English language also needs small correction. For example, in lines 98, 188 etc.

Author Response

Replies to the comments of reviewer #2

Comment: As presented results could be used for the Ti powders production, would be very useful to add some information or preliminary estimations in Introduction on process economy in comparison to existing Ti powders production methods. In other words is proposed approach not too expensive for example in comparison of Ti powders production using powder metallurgy processes?

Answer:

- HDH is the cheapest method for manufacturing Ti powder. Therefore, we added a new reference and following sentence in the introduction part.

 Ref. [9] : Fang, Z.Z.; Paramore, J.D.; Sun, P.; Chandran, K.S.R.; Zhang, Y.; Xia, Y.; Cao, F.; Koopman, M.; Free, M. Powder metallurgy of titanium – past, present, and future. Inter. Mater. Rev. 2018, 63, 407-459, doi.org/10.1080/09506608.2017.1366003.

“The HDH method is a technique that can produce a large amount of powders in an inexpensive method because the powders are fabricated by crushing after hydrogenating Ti [9].” (line 43)

Comment: Figure 1 (a) presents Ti powders fabricated using HDH process. Would be useful to add TiH2 powders SEM views too.

Answer:

- As mentioned by the reviewer, we added SEM image of TiH2 powders in Fig. 1.

Comment: It would be also very useful to add XRD results Ti and TiH2 powders showing phase transformation during HDH process and to confirm that all the hydrogen was released in final Ti powders after HDH.

Answer:

- As mentioned by the reviewer, we added XRD results of the initial Ti sponge, the hydrogenated TiH2 powder, and the dehydrogenation Ti powder in Fig. 2. Also we added following sentences.

 “Phase analyses of the samples were carried out using an X-ray diffractometer (XRD) with Cu Kα radiation in the 2θ range of 30-80°.” (line 78)

 “Fig. 2 shows the XRD patterns of the initial Ti sponge, the hydrogenated TiH2 powder, and dehydrogenated Ti powder. Initially, the Ti sponge had α-Ti phase with a hexagonal close packed structure, and after the hydrogenation annealing, it was transformed to TiH2 phase. Thereafter, brittle TiH2 was pulverized to powders and hydrogen was removed through annealing in a vacuum atmosphere. The XRD result after the dehydrogenation annealing confirmed that all hydrogen was removed and the crystal structure was returned to the original α-Ti phase.” (line 121)

Comment: English language also needs small correction. For example, in lines 98, 188 etc.

Answer:

- As mentioned by the reviewer, we carefully checked English.

“Fig. 1. Morphology of (a) TiH2 and (b) Ti powders fabricated by HDH process and (c) size distribution of TiH2 and Ti powders.” (line 108)

“The oxygen concentration of TiH2 was increased after the ball milling process and TEM and XPS results showed that the TiO2 and TiO oxide layer with the thickness of 14 nm was formed on the surface of the TiH2 powders.” (line 213)

“The decrease in the oxygen concentration could be achieved when the atomic hydrogen was released on the surface during the dehydrogenation process.” (line 216)

Reviewer 3 Report

This paper investigates the oxygen reduction by atomic hydrogen during the dehydrogenation process of titanium hydride powders. A detailed study is performed, with careful characterizations and reasonable analyzes. A few minor changes should be made before publication:

 1, Line 34-43, authors need introduce more reference on the use of HDH Ti powders, rather than EIGA Ti powders, and clearly demonstrate the difference of HDH Ti powders and EIGA Ti powders, such as particle morphology, powder flowing, interstitial contents and so on.

2, Line, 162-168, authors need consider the removal of adsorbed water from the surface of TiH2 powder during heating in Fig. 6.

Author Response

Replies to the comments of reviewer #3

Comment: Line 34-43, authors need introduce more reference on the use of HDH Ti powders, rather than EIGA Ti powders, and clearly demonstrate the difference of HDH Ti powders and EIGA Ti powders, such as particle morphology, powder flowing, interstitial contents and so on.

Answer:

- As mentioned by the reviewer, we added 3 new references for HDH Ti powders. Also, to explain the difference of HDH Ti powders and EIGA Ti powders we added following sentences.

Ref. [9] : Fang, Z.Z.; Paramore, J.D.; Sun, P.; Chandran, K.S.R.; Zhang, Y.; Xia, Y.; Cao, F.; Koopman, M.; Free, M. Powder metallurgy of titanium – past, present, and future. Inter. Mater. Rev. 2018, 63, 407-459, doi.org/10.1080/09506608.2017.1366003.

Ref. [10] : Azevedo, C.R.F.; Rodrigues, D.; Neto, F.B. Ti–Al–V powder metallurgy (PM) via the hydrogenation–dehydrogenation (HDH) process. JALCOM 2003, 353, 217-227, doi.org/10.1016/S0925-8388(02)01297-5.

Ref. [11] : Baril, E.; Lefebvre, L.P.; Thomas, Y. Interstitial elements in titanium powder metallurgy: sources and control. Powder Metall. 2011, 54, 183-186, doi:10.1179/174329011x13045076771759.

“The Ti powder fabricated by EIGA has a spherical shape, whereas the Ti powder fabricated by HDH method has an irregular shape because it is fabricated by grinding [9,10]. Thus, the powder flowability of the HDH Ti powder is much lower than that of the EIGA Ti powder. In addition, in the case of HDH Ti powder, the oxygen concentration is higher than EIGA Ti powder due to the contamination during milling [11], and it is necessary to develop a technology for reducing the oxygen concentration.” (line 46)

Comment: Line, 162-168, authors need consider the removal of adsorbed water from the surface of TiH2 powder during heating in Fig. 6.

Answer:

- GD-MS data cannot distinguish between the amount of H2O generated by the evaporation of water adsorbed on the surface and that generated by the reduction of the oxide layer. Therefore, in Fig. 6, the H2O flow due to evaporation of water adsorbed on the surface cannot be removed. However, according to the reviewer's comment, we added the following sentences.

 “H2O flow is detected in GD-MS during heating metal materials when H2O adsorbed on the metal surface evaporates or the oxide layer on the surface is reduced by hydrogen. Evaporation of H2O adsorbed on the surface would occur near the boiling point of water and the generation of H2O at higher temperatures would be caused by the reduction of the oxide layer.” (line 192)

Round 2

Reviewer 2 Report

Line 183: “….gibbs free energy..” change to “…Gibbs free energy…” and everywhere “..gibbs..” change to “…Gibbs…”.

Line 215: “…increase inH2O flow…” change to “…increase in H2O flow…”.

Author Response

As mentioned by the reviewer, we changed follows.

- ‘gibbs’ ‘Gibbs’

- ‘inH2O’ ‘in H2O’
